# Shapeshifter: a Parameter-efficient Transformer using Factorized Reshaped Matrices

**Aliakbar Panahi**[1,2,*]  **Seyran Saeedi**[1,3,*]  **Tom Arodz**[1,†]

[1] Department of Computer Science, Virginia Commonwealth University, Richmond, VA
[2] C3 AI, Redwood City, CA
[3] Dept. of Electrical and Computer Engineering, University of California, Santa Barbara, CA

ali.panahi@c3.ai    seyran@ucsb.edu    tarodz@vcu.edu

## Abstract

Language models employ a very large number of trainable parameters. Despite being highly overparameterized, these networks often achieve good out-of-sample test performance on the original task and easily fine-tune to related tasks. Recent observations involving, for example, intrinsic dimension of the objective landscape and the lottery ticket hypothesis, indicate that often training actively involves only a small fraction of the parameter space. Thus, a question remains how large a parameter space needs to be in the first place — the evidence from recent work on model compression, parameter sharing, factorized representations, and knowledge distillation increasingly shows that models can be made much smaller and still perform well. Here, we focus on factorized representations of matrices that underpin dense, embedding, and self-attention layers. We use low-rank factorized representation of a reshaped and rearranged original matrix to achieve space efficient and expressive linear layers. We prove that stacking such low-rank layers increases their expressiveness, providing theoretical understanding for their effectiveness in deep networks. In Transformer models, our approach leads to more than tenfold reduction in the number of total trainable parameters, including embedding, attention, and feed-forward layers, with little degradation in on-task performance. The approach operates out-of-the-box, replacing each parameter matrix with its compact equivalent while maintaining the architecture of the network.

## 1 Introduction

Natural language models involve large number of parameters. A single encoder-decoder Transformer [1] in its base variant has about 44 million parameters, not counting the word embedding matrix, which adds another 10 million or more, depending on the chosen vocabulary or tokenization scheme. Base variant of encoder-only BERT [2], including the embedding, has about 108 million parameters. GPT-3 [3] has about 175 billion parameters, and the largest of the Switch Transformer [4] models has 1.5 trillion. This explosion in the model size has led to increased interest in approaches for reducing the number of parameters in the model.

Models with high-dimensional parameter space have much lower intrinsic dimension [5], that is, training trajectory can be successfully restricted to a random, smaller-dimensional subspace, even

---

[*]Work performed while at Virginia Commonwealth University.
[†]Corresponding author.

35th Conference on Neural Information Processing Systems (NeurIPS 2021).

though training a small-parameter architecture is often less successful. These observations have been recently extended to the fine-tuning trajectories of language models [6]. Lottery ticket hypothesis [7, 8], recently demonstrated to hold also for language model fine-tuning [9, 10], shows that smaller subnetworks can be selected from a large model, re-trained in isolation, and perform as well as the large model; what those subnetworks are is not known a prior, in absence of the trained large model, though. Some approaches for reducing model size build on this observation to train a smaller model based on an existing large model, for example a 66 million parameter DistillBERT [11] student has been distilled from 108 million parameter BERT-base [2] teacher with little loss in quality. Alternatively, distillation has been used, for example in the context of machine translation, to increase the quality of the models while maintaining their size [12, 13]. Other approaches train a reduced-parameter model de novo, without relying on an already trained large model. For example, DeLighT [14] uses an alternative parameterization of the multi-headed self-attention based on group linear transform to reduce a 62 million parameter Transformer to 22 million.

One simple way to reduce model size involves factorized matrix representations. ALBERT [15] employs a rank $r$ decomposition of a $d \times n_{vocab}$ embedding matrix storing $d$-dimensional embedding vectors for each of the $n_{vocab}$ tokens by using a stack of two linear layers, $d \times r$ on top of $r \times n_{vocab}$. Similar low-rank decomposition is also used implicitly in the multi-headed self-attention in the generic Transformer [1] with hidden dimension $d$ and $n_{heads}$ self-attention heads. In each Transformer head, a $r = d/n_{heads}$-rank factorized representation involving $d \times d/n_{heads}$ key ($K$) and query ($Q$) matrices are used, with the pairwise self-attention scores for sequence $x$ calculated using $x^T K^T Q x$, instead of $x^T W x$ involving a full general attention $d \times d$ trainable matrix $W$ as originally considered in trainable-attention encoder-decoder LSTM models [16]. In both cases, the models are trained from a random initialization of the factorized parameter space, instead of attempting to find the lowest-error factorized representation of an already trained original model.

We explore here a Transformer model that uses an alternative way to decompose a matrix into two smaller matrices. Instead of standard low-rank factorization as above, it involves reshaping and reordering matrix dimensions prior to the decomposition, and is equivalent to a sum of Kronecker products with an efficient implementation. The compact model based on the smaller matrices is trained de novo, and matrices in the compact model are not aimed to be approximations of the matrices in the original model. For non-square model matrices, the approach allows for increased reduction in parameters for the same decomposition rank. For square matrices, the benefits come from increased expressiveness of the decomposition for the same rank, allowing for reducing the rank needed to preserve model accuracy, and thus reducing the model size. Our main contribution is proving that stacking multiple linear layers decomposed this way increases the expressiveness of the network, unlike stacking multiple low-rank layers factorized in the standard way, which can only map into a subspace of dimensionality equal to the rank. Empirically, we show that the decomposition can reduce the size of a simple encoder-decoder Transformer to as little as 4 million parameters, including 2 million for the model and 2 million for the embeddings. The technique can be employed automatically to any matrix, including embedding, dense, attention, and output layers, without requiring any modification of the model architecture.

## 2 Related Work

The two approaches most closely related to ours are word2ket embeddings [17] and parameterized hypercomplex multiplication (PHM) linear layers [18]. Word2ket represents input embedding matrix in language models via a sum of Kronecker products, each involving two smaller matrices. It achieves above 100-fold reduction in the number of trainable embedding parameters; however, their approach relies on lazy tensors to efficiently extract embeddings for a limited number of words in the input sequence, and does not extend to output embedding matrix or other model matrices that perform general linear transformations of their input vectors. PHM linear layers [18] arise from generalizing hypercomplex number layers [19] to arbitrary dimensionality $r$, with the arithmetic over the numbers learned during training. For a given dimensionality $r$, the arithmetic takes form of a sum of $r$ Kronecker products. Irrespective of the original matrix size $n \times m$, each Kronecker product involves a small $r \times r$ matrix and a larger $n/r \times m/r$ matrix, which together have $r^3 + nm/r$ trainable parameters. For $r$ used in practice, the second term dominates, resulting in $r$-fold reduction in size. PHM layers with $r$ as large as 16 have been demonstrated to be effective, leading to a reduction of a 44 million parameter Transformer to a 2.9 million parameter PHM-Transformer. These numbers do

not include the embedding matrix for token embeddings which is not reduced in size by the PHM approach; and which for example for $32,000$ BPE tokens requires about 16 million parameters.

Both of the Kronecker-based compact language model approaches [17, 18], as well as Kronecker-based convolutional [20] and recurrent networks [21], demonstrate that a sum of Kronecker products leads to very compact representations. Here, we advance theoretical understanding of why this low-rank representation has advantages in terms of expressivity compared to a standard matrix factorization with the same rank. We also provide a more efficient, more general Kronecker-based representation. Unlike word2ket, it can be used in linear transformations anywhere in the model, an unlike PHM it also applies to embedding matrices. To represent an $n \times m$ matrix, our approach requires $2r\sqrt{mn}$ instead of $r^3 + nm/r$ parameters required by PHM. For example, for $r = 16$, the highest used by PHM, for a Transformer with hidden dimension of 512, each $512 \times 2048$ feed-forward matrix in the self-attention head would be reduced from 1 million parameters to 69,632 parameters by PHM, but to 16,384 by our approach.

Kronecker product of two matrices is a concrete-basis representation of a finite-dimensional linear operator defined by a tensor product of two underlying linear operators. Thus, Kronecker-product-based representations, including PHM layers and ours, can be seen as special case of a more general family of higher-order tensor-product-based representations. Kronecker-based representation unfolds a $n \times m$ matrix from two smaller matrices, $n_1 \times m_1$ and $n_2 \times m_2$, with $m = m_1 m_2$ and $n = n_1 n_2$. In tensor-based representations, this is generalized to $m = \prod_{j=1}^{o} m_j$ and $n = \prod_{j=1}^{o} n_j$ for some tensor order $o$, leading to unfolding the $n \times m$ matrix from an $o$-way tensor that is then decomposed using low rank leading to reduction in parameters. For example, the tensor-train representation [22] represents a matrix using a series of core tensors with matching ranks. Tensor decomposition has been used successfully in convolutional networks [23, 24] and recurrent networks [25]. In the context of language models, tensor-train representation has been used to construct tensorised embeddings [26], which use order-three tensor representation to obtain 60-fold reduction in embedding in a sequence-to-sequence Transformer, and also use order-six tensors to achieve almost 400-fold reduction in an LSTM model for sentiment analysis. Similarly, word2ket [17] uses order-four tensor representation to achieve more than 90,000-fold reduction in the embedding size for question-answering LSTM model. Beyond embeddings, Tensorized Transformer [27] uses block-term tensor decomposition – a combination of CP and Tucker decompositions – to reduce the multi-headed self-attention layer with $n_{heads}$ by a factor $1/n_{heads}$, leaving other layers intact; this allowed for reducing a 52M Transformer down to 21M trainable parameters, with little drop in BLEU scores on WMT'16 En-De translation task compared to original Transformer. So far, none of the Kronecker- or tensor-product-based approaches have been applied comprehensively to all components of a Transformer model – embeddings, attention, and feed-forward layers.

## 3 Limitations of Factorized Matrices in Deep Networks

A simple approach for reducing the size and complexity of a model involves replacing each linear transformation $n \times m$ matrix $W$ with its factorized, low-rank representation involving an $n \times r$ matrix $A$ and an $r \times m$ matrix $B$, with the rank $r \ll m, n$. During training and inference, the product $AB$ is used instead of $W$, essentially replacing $W$ with a two linear layers, $A$ stacked on $B$. In this way, the number of parameters reduces from $nm$ needed to represent $W$ to $r(n + m)$ needed to represent $A$ and $B$.

One potential approach to use the factorized representation is to take an already trained model, and factorize each matrix $W_j$ into its low-rank approximation $\widehat{W_j} = A_j B_j$ using a method such as SVD. However, analyzing an idealized deep model with linear activations shows that errors from the layers add up [28], $\|\prod_j W_j - \prod_j \widehat{W_j}\| = \sum_j \|W_j - \widehat{W_j}\|$. Instead, a typical approach, used also here, is to train a model de novo in its factorized form, using randomly initialized matrices $A_j$ and $B_j$, without attempting to approximate individual linear transformations of the non-factorized model.

Representing an $n \times m$ matrix $W$ as a low-rank product $AB$ limits the linear transformation – the output is constrained to be a low-dimensional subspace of $\mathbb{R}^n$. Stacking rank-$r$ factorized linear transformations into an idealized deep network with linear activations does not improve the expressiveness, the dimensionality of the space of possible outputs is still constrained by the smallest of the decomposed layers rank. While this may have regularizing effect for medium values of the rank, it may prevent from using very small ranks and thus from creating compact models.

# 4 Stacked Kronecker Product-based Representations

For $n = m$, the standard factorized $(n \times r)\,(r \times m) \to n \times m$ representation is optimal among pairwise-product-based representations in the sense that all elements of a column of $A$ are multiplied with each element of a row of $B$. For non-square matrices, for example matrix $A$ is smaller than $B$, a more efficient all-pairs product representation can be made by increasing the size of $A$ and decreasing the size of $B$.

In Shapeshifter, instead of the standard $(n \times r)\,(r \times m) \to n \times m$ factorized representation, we rearrange the $mn$ parameters of the original matrix $W$ into a $\sqrt{mn} \times \sqrt{mn}$ matrix[3] $W'$, and then use a factorized representation of $W'$ as a $\sqrt{mn} \times r$ matrix $A$ and an $r \times \sqrt{mn}$ matrix $B$:

$$\left(\sqrt{nm} \times r\right)\left(r \times \sqrt{nm}\right) \xrightarrow{\text{multiply}} \sqrt{nm} \times \sqrt{nm} \xrightarrow{\text{reshape}} \sqrt{n} \times \sqrt{m} \times \sqrt{n} \times \sqrt{m}$$
$$\xrightarrow{\text{transpose}} \sqrt{n} \times \sqrt{n} \times \sqrt{m} \times \sqrt{m} \xrightarrow{\text{reshape}} n \times m$$

This representation uses $2r\sqrt{mn}$ parameters instead of $nm$.

We apply this representation to all matrices in attention and feed-forward layers in all multi-head attention blocks, and to the embedding matrices. For non-square matrices, the representation results in parameter saving compared to standard factorization with the same rank $r$. For square matrices, no parameter saving is achieved. However, as we show below, the way the matrix is decomposed allows for increased expressive power.

For $r = 1$, the $n \times m$ matrix $W$ arises from taking products of each element from $\sqrt{nm} \times 1$ matrix $A$ with each element from $1 \times \sqrt{nm}$ matrix $B$. It is known to be equivalent [29] to a Kronecker product $A \otimes B$ involving $\sqrt{n} \times \sqrt{m}$ matrices $A$, $B$ resulting from reshaping the single-column and single-row matrices, respectively. Beyond $r = 1$, the representation is equivalent to representing $W$ as sum of $r$ Kronecker products, $W = \sum_{j=1}^{r} A_j \otimes B_j$. Compared to the equivalent sum of Kronecker-products representation , using the matrix factorization representation avoids the explicit summation of $r$ terms. In current libraries, it results in a simpler implementation involving only matrix multiplication, reshaping and transposing, and is more efficient compared to performing individual Kronecker products followed by an explicit summation along the rank dimension.

## 4.1 Expressiveness of stacked Kronecker-product Layers

Certain types of matrices can easily be expressed as a sum of Kronecker products, but not in a $r$-rank factorized matrix form. For example, identity mapping on $\mathbb{R}^n$ does not admit low-rank factorization, but it can be decomposed as a single Kronecker product of two smaller identity matrices. More broadly, the output of a sum of Kronecker products layer is not limited to $r$-dimensional subspace; a Kronecker product of two orthogonal matrices is an orthogonal matrix [30], thus even a single $A \otimes B$ term can model isomorphisms $\mathbb{R}^n \to \mathbb{R}^n$. The converse is not true, most matrices require Kronecker representation with a high rank, and the question of approximating various types of matrices with a sum of Kronecker products has received ample attention [31, 32, 30, 33].

Here, we focus on a different question – how does the expressive power of a stack of layers, each involving a sum of Kronecker products instead of generic linear transformation, grow with the network depth. We analyze an idealized network formed by a stack of Kronecker layers with linear activations. Given high-enough rank, a single layer is enough to represent any matrix. We first analyze what the depth needs to be if rank is restricted to two, and then investigate how quickly the depth can be reduced as the rank increases.

Below, we use the following notation. Support $\text{supp}(x)$ of vector $x$ is a set of indices at which $x$ is not null. We use $[n] = \{1, ..., n\}$. $I$ is identity matrix. $\mathbb{1}_{jk}$ is a matrix with unity at row $j$ and column $k$ and null elsewhere; $\mathbb{1}_k$ is a shorthand for $\mathbb{1}_{kk}$. To avoid ambiguity, we use square brackets for individual entries of a matrix or a vector; thus $A[j, :]$ represents $j$-th row of matrix $A$, while $A_j$ represents $j$-th matrix out of a series, as used above. For index set $Z$ of cardinality $k$ and a square matrix $A$, by $A[Z]$ we represent a $k \times k$ submatrix of $A$ obtained by taking the intersection of columns and rows from set $Z$.

We first define $k$-variant matrices, which will be a tool in exploring stacked layers.

---

[3]If $\sqrt{mn}$ is not an integer, we round it up, and use a submatrix as representation of $W$.

**Definition 1** (*k*-variant matrix). *For an $n \times n$ orthogonal matrix $U$ and an index set $Z \subset [n]$, by writing $U^Z$ we indicate that both $U$ and its inverse $U^T$ act as identity on all the dimensions not in set $Z$. That is, for each $x \in \mathbb{R}^n$, $\mathrm{supp}(Ux - x) \subseteq Z$ and $\mathrm{supp}(U^T x - x) \subseteq Z$. If $Z$ has cardinality at most $k$, we call $U^Z$ a k-variant matrix.*

The notion of $k$-variant matrices generalizes Givens rotation matrices [34] used for example in QR decomposition. A Givens rotation in $n$ dimensions is any rotation that acts on a plane spanned by two of the $n$ coordinate axes – it is thus 2-variant. QR decomposition can be used to show that each matrix can be represented as a product of 2-variant matrices. We will then show that 2-variant matrices, or more broadly, $k$-variant matrices for $k \leq \sqrt{n}$, can be represented using sum of Kronecker products.

**Theorem 2** (Decomposition into layers involving Kronecker products). *Let $n = m^2$ for some $m \in \mathbb{N}$. Any orthogonal $n \times n$ matrix $U$ can be represented as*

$$U = \prod_{i=1}^{L} \sum_{j=1}^{2} A_{ij} \otimes B_{ij},$$

*where $A_{ij}$ and $B_{ij}$ are $\sqrt{n} \times \sqrt{n}$ matrices, and $L = O\left(n^2\right)$.*

*Proof.* QR decomposition using Givens matrices will result in $Q$ being a product of up to $n(n-1)/2$ Givens rotations, which are 2-variant, and $R$ an upper triangular matrix which for orthogonal matrices is a diagonal matrix with $+1$ and $-1$ elements. The sign of any pair of $-1$ diagonal entries can be negated by a 2-variant matrix; at most $n/2$ such matrices are needed to convert the diagonal into identity, since there are at most $n - 1$ negative elements in $R$. In total, up to $n^2/2$ 2-variant matrices are needed to represent any given orthogonal matrix.

Next, we set to show that any 2-variant matrix can be represented as a product of at most three matrices, each of the form $A_1 \otimes B_1 + A_2 \otimes B_2$, using $m \times m$ matrices.

Let $U^{\{k,q\}}$ be an arbitrary 2-variant orthogonal $m^2 \times m^2$ matrix for arbitrary $k, q \in [m^2]$. That is, $y = U^{\{k,q\}}x$ acts only on $x[k]$ and $x[q]$ to produce $y[k]$ and $y[q]$, for $l \neq k, q$ we have $y[l] = x[l]$.

Consider two vectors, $x_A, x_B \in \mathbb{R}^m$. Kronecker product of $x = x_A \otimes x_B$ of these two vectors is a vector $x \in \mathbb{R}^n$ with entries defined through products

$$
\begin{aligned}
x[k] &= x_A[\kappa]x_B[\lambda] &&\text{for}\quad k = (\kappa - 1)m + \lambda, \\
x[q] &= x_A[\pi]x_B[\rho] &&\text{for}\quad q = (\pi - 1)m + \rho
\end{aligned}
$$

for $\kappa, \lambda, \pi, \rho \in [m]$.

A matrix $A \otimes B$ acts on vectors $x = x_A \otimes x_B$ as $(A \otimes B)x = (Ax_A) \otimes (Bx_B)$. For example, let $B^{\{\lambda,\rho\}}$ be a 2-variant matrix action on $\lambda, \rho$, then $\mathbb{1}_\nu \otimes B^{\{\lambda,\rho\}}$ is a acting on dimensions $k = (\nu - 1)m + \lambda$ and $q = (\nu - 1)m + \rho$ in the same way as $B^{\{\lambda,\rho\}}$ acts on dimensions $\lambda$ and $\rho$, and producing either identity or null elsewhere, as shown below:

$$
\begin{bmatrix} 0 & & \\ & 1 & \\ & & 0 \end{bmatrix} \otimes \begin{bmatrix} \alpha & & -\beta \\ & 1 & \\ \beta & & \alpha \end{bmatrix} = \begin{bmatrix} 0 & & & & & & \\ & \ddots & & & & & \\ & & \alpha & & -\beta & & \\ & & & 1 & & & \\ & & \beta & & \alpha & & \\ & & & & & \ddots & \\ & & & & & & 0 \end{bmatrix}.
$$

First, consider $k, q$ such that $\kappa = \pi = \nu$ for some $\nu \in [m]$. Define a 2-variant $B^{\{\lambda,\rho\}}$ such that $B[\{\lambda, \rho\}] = U[\{k, q\}]$, that is, $B$ acts on $\lambda, \rho$ in the same way as $U$ acts on $k, q$. We then have

$$U^{\{k,q\}} = \mathbb{1}_\nu \otimes B^{\{\lambda,\rho\}} + (I - \mathbb{1}_\nu) \otimes I.$$

A symmetric construction holds if $\lambda = \rho = \nu$. In both cases, we can represent a 2-variant orthogonal matrix by $A_1 \otimes B_1 + A_2 \otimes B_2$.

The general case of $U^{\{k,q\}}$ for $\kappa \neq \pi$ and $\lambda \neq \rho$ can be addressed by converting it into the case $\kappa = \pi$ via transforming $\kappa$ into $\pi$, applying the special case, and then transforming $\pi$ back into $\kappa$.

We show that an orthogonal matrix $V$ of the form $V = C \otimes D + C' \otimes D'$ exists such that

$$U^{\{k,q\}} = V^T \left( \mathbb{1}_\pi \otimes B^{\{\lambda,\rho\}} + (I - \mathbb{1}_\nu) \otimes I \right) V,$$

where $B[\{\lambda,\rho\}] = U[\{k,q\}]$, the $\lambda, \rho$ columns/rows of $B$ are formed by taking submatrix of $U$ defined by columns/rows $k, q$.

To construct $V$, set $C = P_{\kappa \to \pi}$, a permutation matrix that moves $\kappa$ to $\pi$, and $D = \mathbb{1}_\lambda$. Also set $C' = I$ and $D' = I - \mathbb{1}_\lambda$. Columns of $C \otimes D$ and $C' \otimes D$ are linearly independent, and each of the two matrices is orthogonal, thus $V$ is orthogonal.

Consider $x = x_A \otimes x_B$, we then have

$$x' = Vx = P_{\kappa \to \pi} x_A \otimes \mathbb{1}_\lambda x_B + I x_A \otimes (I - \mathbb{1}_\lambda) x_B.$$

The first term will use the permutation to move $x_A[\kappa]$ to position $\pi$, where it will be multiplied by $x_B[\lambda]$, and placed at $x'[k']$ for $k' = (\pi - 1)m + \lambda$; all other entries $x_A[\cdot]x_B[\lambda]$ will be rearranged as well by the permutation. For $m = 3$, $\lambda = 2$, $\kappa = 1$, $\pi = 2$, the transformation that moves $x[k = 2] = x[(\kappa - 1)m + \lambda] = \alpha_1 \beta_2$ into $x'[k' = 5] = x'[(\pi - 1)m + \lambda]$ and thus brings it into position ready for the special case, is illustrated below

$$\begin{bmatrix} 0 & 1 & 0 \\ 1 & 0 & 0 \\ 0 & 0 & 1 \end{bmatrix} \begin{bmatrix} \alpha_1 \\ \alpha_2 \\ \alpha_3 \end{bmatrix} \otimes \begin{bmatrix} 0 & 0 & 0 \\ 0 & 1 & 0 \\ 0 & 0 & 0 \end{bmatrix} \begin{bmatrix} \beta_1 \\ \beta_2 \\ \beta_3 \end{bmatrix} = \begin{bmatrix} \alpha_2 \\ \alpha_1 \\ \alpha_3 \end{bmatrix} \otimes \begin{bmatrix} 0 \\ \beta_2 \\ 0 \end{bmatrix} =$$

$$= \begin{bmatrix} 0 & \alpha_2\beta_2 & 0 & 0 & \alpha_1\beta_2 & 0 & 0 & \alpha_3\beta_2 & 0 \end{bmatrix}^T.$$

The second term will preserve $x_A[\cdot]x_B[\cdot]$ for all entries of $x_B$ except $\lambda$; this includes $\rho$, since in the general case $\lambda \neq \rho$. The $k, q$ entries of $x$ will be now at $k', q$ entries of $x'$, and applying $\mathbb{1}_\pi \otimes B^{\{\lambda,\rho\}} + (I - \mathbb{1}_\nu) \otimes I$ on $x'$ will be equivalent to acting on positions $k, q$ of $x$. The inverse orthogonal transformation $V^T$ will restore the transformed values at positions $k', q$, as well as all other entries $x_A[\cdot]x_B[\lambda]$, back to their original positions. The transformation $V^T$ is of the same form as $V$, except using $P_{\kappa \to \pi}^{-1}$ instead of $P_{\kappa \to \pi}$, since the other matrices involved in $V$ are diagonal with unit-magnitude entries. In summary, in the general case, $m \times m$ matrices $A_{ij}$ and $B_{ij}$ exist such that any 2-variant orthogonal matrix can be represented as $\prod_{i=1}^3 \sum_{j=1}^2 A_{ij} \otimes B_{ij}$.

Together, any orthogonal $n = m^2$ dimensional matrix $U$ can be decomposed into $U = \prod_{i=1}^{3n^2/2} \sum_{j=1}^2 A_{ij} \otimes B_{ij}$. $\qquad\qquad\square$

**Corollary 3.** *Any linear $n \times m$ matrix $W$ can be represented as a product of a finite sequence of sums of two Kronecker products of $\lceil n \rceil \times \lceil m \rceil$ matrices followed by selecting an $n \times m$ submatrix.*

*Proof.* The result immediately follows by considering an orthogonal matrix with dimension $\max(\lceil n \rceil^2 \times \lceil m \rceil^2)$, observing that an arbitrary, possibly rank-deficient diagonal matrix $D$ can be decomposed as a product of a series of matrices of the form $\mathbb{1}_k \otimes (I - (D[j,k] - 1)\mathbb{1}_j) + (I - \mathbb{1}_k) \otimes I$, one per diagonal entry $D[j,k]$, which allows for producing arbitrary rank-deficient square matrices, from which non-square matrices can be selected. $\qquad\qquad\square$

The above theorem shows that stacking layers represented compactly as $(\sqrt{nm} \times r)(r \times \sqrt{nm})$ for $r = 2$ increases the expressive power of the network. In the idealized case with linear activation function, adding such layers allows the network to eventually produce arbitrary $\mathbb{R}^m \to \mathbb{R}^n$ linear transformations, unlike stacking standard factorized layers $(n \times r)(r \times m)$, which always maps into an $r$-dimensional subspace irrespective of the number of layers.

With rank $r = 2$, we may need as many as $3n^2/2$ layers to represent an orthogonal $n \times n$ matrix. On the other extreme, only one layer is needed if the rank is $n$, since arbitrary $n \times n$ matrix can be seen as a concatenation of $n$ tiles of size $\sqrt{n} \times \sqrt{n}$. As long as we define $n$ indicator $\sqrt{n} \times \sqrt{n}$ matrices $\mathbb{1}_{jk}$ and use them as tile position selectors, we can use a sum of Kronecker products of the selection with the corresponding tile to produce the matrix. In between these extremes, the number of layers needed to increase the expressiveness of the network to allow arbitrary transformations depends on the rank $r$ in the Shapeshifter representation in the following way.

**Theorem 4** (Depth-rank tradeoff for deep models involving Kronecker products). *Let $n = m^2$ for some $m \in \mathbb{N}$, and let $r \leq m$. Any orthogonal $n \times n$ matrix $U$ can be represented as a product of a sequence of sums of two Kronecker products of $m \times m$ matrices,*

$$U = \prod_{i=1}^{L} \sum_{j=1}^{r} A_{ij} \otimes B_{ij},$$

*where $A_{ij}$ and $B_{ij}$ are $\sqrt{n} \times \sqrt{n}$ matrices, and $L = O\left(n^2/r\right)$.*

*Proof.* Instead of 2-variant orthogonal matrices, we use $r$-variant ones. Consider $r = 3$, with $U^{\{k,q,t\}}$, and with $k = (\kappa - 1)m + \lambda$, $q = (\pi - 1)m + \rho$, and $t = (\tau - 1)m + \upsilon$. We now have two groups of indices, $\kappa, \pi, \tau$ and $\lambda, \rho, \upsilon$. If $\kappa = \pi = \tau = \nu$ for some $\nu$, we have a special case as in Theorem 2

$$U^{\{k,q,s\}} = \mathbb{1}_\nu \otimes B^{\{\lambda,\rho,\upsilon\}} + (I - \mathbb{1}_\nu) \otimes I;$$

we have a similar special case if $\lambda = \rho = \upsilon$.

The general case involves no equality in either of the two groups. Then, we can proceed similarly as for 2-variant general case, but we need a permutation matrix $P_{\kappa \to \tau}$ and separately a permutation matrix $P_{\pi \to \tau}$, which allow us to define

$$x' = Vx = P_{\kappa \to \tau} x_A \otimes \mathbb{1}_\lambda x_B + P_{\pi \to \tau} x_A \otimes \mathbb{1}_\rho x_B + I x_A \otimes (I - \mathbb{1}_\lambda - \mathbb{1}_\rho) x_B.$$

Here, $V$ is an orthogonal transformation that uses a sum of three Kronecker products as a transformation that maps the general case to the $\kappa = \pi = \tau$ case. The construction naturally extends into higher $r$ as long as $r \leq m$, the number of diagonal entries in matrices $A$: we need $V$ to be a sum of $r$ Kronecker products involving distinct dimensions of $A$. In summary, we need less than $\prod_{i=1}^{3} \sum_{j=1}^{r} A_{ij} \otimes B_{ij}$ to represent arbitrary $r$-variant orthogonal matrix. A product of two $k$-variant matrices is at most $2k$-variant. Grouping $r/2$ Givens rotations or, more generally, 2-variant matrices can result in at most a $r$-variant matrix, thus we need at most $L = O\left(3n^2/r\right)$ layers, each involving a Kronecker product of rank at most $r$. $\qquad \square$

As is common in theoretical work on deep networks, the results are limited to idealized multilayer models with linear activations. However, they can provide insights into networks with nonlinearities; for example, activations such as ReLU that are linear on large part of their input will lead to piece-wise linear network, and the results above hold in a piece-wise fashion.

## 5 Experimental Results

We validated the approach on machine translation using sequence-to-sequence models. The experiments were performed on a single V100 (longest run: 6 days) or A100 GPU (longest run: 1 day). Time overhead experiments were performed on a dedicated workstation with a single NVIDIA RTX 3090 GPU. As the basis for Shapeshifter, we used Transformer [1], with embedding and encoder/decoder layers replaced with compact representations. In Shapeshifter, we used small rank for all matrices in the multi-head self-attention blocks. For embedding matrices, which are much larger and can be reduced more effectively, we used higher rank, while aiming to keep the total size of factorized embeddings below the total size of the rest of the factorized encoder/decoder. The time overhead

Table 1: **Overhead introduced by Shapeshifter compact representation.** We compared T5-small Transformer model [35] with a Shapeshifter model resulting from the T5-Small model, with rank 256 for embedding and rank 16 within the multi-headed self-attention blocks. We measured time to train 10 epochs, and time to perform translations using a trained model on the full test set.

| Dataset | Training time [s] | | Inference time [s] | |
|---|---|---|---|---|
| | Transformer | Shapeshifter 16/256 | Transformer | Shapeshifter 16/256 |
| En-Ro [36] | 16,192 | 18,007 (+11%) | 110 | 124 (+13%) |
| De-En [37] | 2,917 | 3,426 (+17%) | 203 | 219 (+8%) |

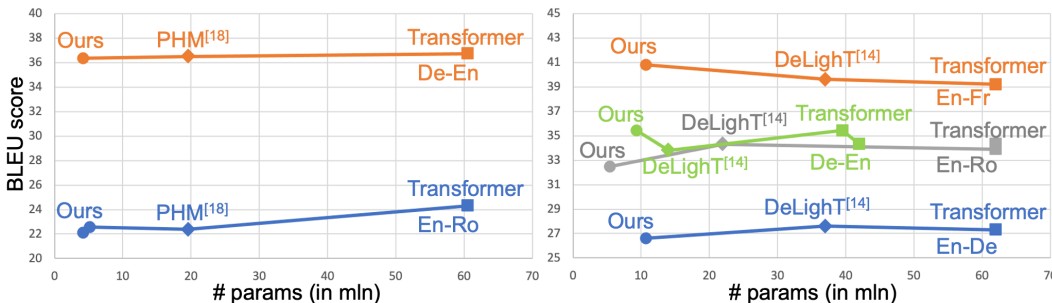

Figure 1: Comparison of the proposed Shapeshifter compact representation applied to Transformer with two alternative approaches for compact Transformers: PHM layers [18] and DeLighT [14]. See Tables 2 and 3 for details.

(see Table 1) introduced by the representation is modest, up to 17% increase in running time during training, and up to 13% during predictions with a trained model.

We compared our approach to two recently proposed approaches for reducing model size: DeLighT [14] and PHM Layers [18]. DeLighT uses separate vocabulary for the source language encoder embeddings, and a different vocabulary for the target language decoder embedding and output layer. PHM Layers, like its predecessor Quaternion Transformer [19], uses the same Byte-Pair Encoding [38] in both encoder and decoder. To facilitate comparisons, we followed both approaches and trained two Shapeshifter variants, one with separate embeddings and one with single embedding.

## 5.1 Comparison with PHM Layers

In comparisons with PHM layers [18], we used T5-small Transformer [35], which shares a single embedding matrix in both the encoder and the decoder. We use Huggingface Transformers [40] for PyTorch [41] implementation. We evaluate the approach on IWLSLT'14 German-to-English (De-En) [37] and WMT'18 English-to-Romanian (En-Ro) [36] datasets. For the first dataset we use learning

Table 2: **Results of comparisons with PHM layers [18].** Model performance measured using BLEU [39] on dev set. We also list total number of model parameters, its breakdown into number of parameters in the encoder-decoder multi-headed self-attention and feed-forward layers, and separately in the all the embedding in the model. We also express the parameter reduction as a percentage and as a fold change compared to the size of the corresponding Transformer model. Shapeshifter $r_{encoder}$ / $r_{embedding}$ denotes encoder/decoder and embedding ranks.

| Model | Total | Enc./Dec. | Emb. | Pct. | Fold | BLEU |
|---|---|---|---|---|---|---|
| WMT'18 English-to-Romanian (En-Ro) dataset [36] | | | | | | |
| Transformer [1], from [18] | – | 44M | – | – | – | 22.79 |
| PHM-Tm n = 16 [18] | – | 2.9M | – | – | – | 19.63 |
| *Transformer [1] | 60.5M | 44M | 16.5M | 100.0 | 1 | 24.30 |
| *PHM-Tm n = 16 [18] | 19.6M | 3.2M | 16.5M | 32.4 | 3.08 | 22.38 |
| *Shapeshifter 24/256 | 5.2M | 3.1M | 2.1M | 8.62 | 11.59 | 22.56 |
| *Shapeshifter 16/256 | 4.2M | 2.1M | 2.1M | 6.95 | 14.39 | 22.12 |
| IWSLT'14 German-to-English (De-En) dataset [37] | | | | | | |
| Transformer [1], from [18] | – | 44M | – | – | – | 36.68 |
| PHM-Tm n = 16 [18] | – | 2.9M | – | – | – | 33.89 |
| *Transformer [1] | 60.5M | 44M | 16.5M | 100.0 | 1 | 36.72 |
| *PHM-Tm n = 16 [18] | 19.6M | 3.2M | 16.5M | 32.4 | 3.08 | 36.49 |
| *Shapeshifter 16/256 | 4.2M | 2.1M | 2.1M | 6.95 | 14.39 | 36.36 |

\* denotes our experimental results, other results taken from the referenced paper.
– denotes the information was not available in the referenced paper.

rate 2e-3 with batch size of 128, while for the larger En-Ro dataset we use 3e-3 with batch size of 192. We used LAMB optimizer [42] with inverse square root scheduler, dropout 0.1, no weight decay, and 0.1 label-smoothed cross entropy loss.

To facilitate comparison of our results with PHM Layers approach under the same conditions, we reimplemented the approach and applied it to T5-Small Transformer that we used as basis for experiments with our method. The small increase in size of the PHM Transformer observed in our re-implementation of PHM layers compared to the parameter size provided in the literature [18] (3.2M vs 2.9M parameters) results from the fact that the underlying T5 Transformer that we build on uses three separate $d_{hid} \times d_{hid}$ matrices for self-attention query, key, and value weights for all heads, while the Transformer used in [18] stores all three as a single, concatenated $d_{hid} \times 3d_{hid}$ matrix.

The results presented in Table 2 and in the left panel of Figure **??** indicate that while our full models are four times smaller than full PHM models, they offer very similar on-task performance. The parameter reduction comes not only from factorized embeddings in Shapeshifter, but also from matrices inside the attention blocks in encoder/decoder: both Shapeshifter 16 / 256 and PHM $n = 16$ use a sum of 16 Kronecker products to represent a matrix, yet PHM results in a 1.5 times larger encoder/decoder when applied to the same exact base architecture.

## 5.2 Comparison with DeLighT

Following DeLighT [14], we used Transformer models as implemented in fairseq [45]. We benchmarked the Shapeshifter models on four datasets: IWSLT'14 German-to-English (De-En) [37], WMT'16 English-to-Romanian (En-Ro) [43], WMT'14 English-to-German (En-De) [44], and WMT'14 English-to-French (En-Fr) [44]. The specific Transformer architecture used by DeLighT, and also in our experiments, varies depending on the dataset. For the WMT'14 En-De we used 12 transformers blocks with 4 attention heads, 512 as the embedding dimension and 1024 for the fan out fully-connected layer. For the other three datasets we used 12 transformers blocks with 8 attention heads, 512 as the embedding dimension and 2048 for the fan out fully-connected layer. For the training setup we have used Adam optimizer, learning rate of 5e-4, inverse square root scheduler with 15K warmup steps, dropout 0.3, weight decay of 1e-4, and 0.1 label-smoothed cross entropy loss.

Compared to the smallest of the DeLighT models available for each dataset, as summarized in Table 3 and in the right panel of Figure **??**, the Shapeshifter models have between 1.5 and 4 times

Table 3: **Results of comparisons with DeLighT [14].** Same format as in Table 2.

| Model | Total | Enc./Dec. | Emb. | Pct. | Fold | BLEU |
|---|---|---|---|---|---|---|
| IWSLT'14 German-to-English (De-En) dataset [37] | | | | | | |
| Transformer [1], from [14] | 42M | – | – | 100.0 | 1 | 34.3 |
| DeLighT [14] | 14M | – | – | 35.5 | 2.82 | 33.8 |
| *Transformer [1] | 39.5M | 31.5M | 7.9M | 100.0 | 1 | 35.43 |
| *Shapeshifter 64/256 | 9.3M | 7.2M | 2.1M | 23.6 | 4.24 | 35.43 |
| WMT'16 English-to-Romanian (En-Ro) dataset [43] | | | | | | |
| Transformer [1], from [14] | 62.0M | 44.1M | 17.9M | 100.0 | 1 | 34.3 |
| DeLighT [14] | 22.0M | – | – | 35.5 | 2.82 | 34.3 |
| *Transformer [1] | 62.0M | 44.1M | 17.9M | 100.0 | 1 | 33.89 |
| *Shapeshifter 64/256 | 10.4M | 8.2M | 2.2M | 16.8 | 6 | 34.55 |
| *Shapeshifter 24/256 | 5.4M | 3.1M | 2.2M | 8.6 | 11.63 | 32.48 |
| WMT'14 English-to-German (En-De) dataset [44] | | | | | | |
| Transformer [1], from [14] | 62.0M | 44.1M | 17.9M | 100.0 | 1 | 27.3 |
| DeLighT [14] | 37M | – | – | 59.6 | 1.68 | 27.6 |
| *Shapeshifter 64/256 | 10.7M | 8.2M | 2.5M | 17.2 | 5.7 | 26.6 |
| WMT'14 English-to-French (En-Fr) dataset [44] | | | | | | |
| Transformer [1], from [14] | 62.0M | 44.1M | 17.9M | 100.0 | 1 | 39.2 |
| DeLighT [14] | 37.0M | – | – | 59.6 | 1.68 | 39.6 |
| *Shapeshifter 64/256 | 10.7M | 8.2M | 2.5M | 17.3 | 5.78 | 40.78 |

fewer parameters, yet offer similar quality; Shapeshifter achieves higher score than DeLighT on two datasets, and lower score on two other datasets.

## 5.3 Comparison with Standard Low-rank Factorization

To validate experimentally the theoretical results from Section 4.1 concerning the increased expressiveness of the factorization type used in Shapeshifter compared to standard $n \times r$, $r \times m$ low-rank factorization, we trained models factorized this way using the same encoder/decoder rank of 16, and the same embedding rank of 256 as in Shapeshifter. To test only the stack of Transformer layers, we also trained models that do not factorize the embedding matrix. We used one small dataset, IWSLT'14 German-to-English, and one large, WMT'18 English-to-Romanian. The results in Table 4 show that the size of the model is reduced substantially in the Kronecker-based approach compared to standard low-rank representation. The reduction comes predominantly from the embedding, which involve non-square matrices. Despite more than two-times larger number of parameters, standard low-rank representation shows much higher loss on the training set, with no significant difference in terms of generalization outside of the training set, confirming that the difference between the two representations results from increased expressiveness of a stack of low-rank Kronecker-product-based layers. Similar behavior can be observed when the comparison involves models where the embedding matrices are not factorized.

Table 4: **Comparison with standard low-rank representation.** Factorized models are trained de novo on a small (IWSLT'14 De-En) and a large (WMT'18 En-Ro) dataset.

| Dataset | Factorization type | Total | Enc/Dec | Emb. | BLEU | Train Loss | Dev Loss |
|---------|--------------------|-------|---------|------|------|------------|----------|
| De-En | Shapeshifter 16 / 256 | 4.2M | 2.1M | 2.1M | 36.36 | 2.59 | 2.67 |
| De-En | Low-rank 16 / 256 | 10.6M | 2.2M | 8.4M | 26.68 | 2.97 | 2.99 |
| De-En | Shapeshifter 16 / – | 18.6M | 2.1M | 16.5M | 35.85 | 2.66 | 2.73 |
| De-En | Low-rank 16 / – | 18.7M | 2.2M | 16.5M | 32.01 | 2.86 | 2.90 |
| En-Ro | Shapeshifter 16 / 256 | 4.2M | 2.1M | 2.1M | 22.12 | 2.19 | 3.09 |
| En-Ro | Low-rank 16 / 256 | 10.6M | 2.2M | 8.4M | 17.47 | 2.32 | 3.37 |
| En-Ro | Shapeshifter 16 / – | 18.6M | 2.1M | 16.5M | 21.93 | 2.16 | 3.09 |
| En-Ro | Low-rank 16 / – | 18.7M | 2.2M | 16.5M | 19.34 | 2.33 | 3.34 |

## 6 Conclusion

Our theoretical and experimental results show that deep models composed of stacked low-rank Kronecker-product-based representations are more expressive, yet smaller, than equivalent models that use standard low-rank factorized matrix representations. These observations help solve a puzzle of how recent methods such as word2ket [17] and PHM layers [18] achieve their impressive parameter reduction rates for embeddings and for encoder/decoder layers, respectively, despite training models de novo, without the help of an existing large model serving as a blueprint for compression or pruning, or as a teacher for knowledge distillation. Building on these results, we provide a comprehensive Shapeshifter[4] approach that works both for the embeddings and for the encoder/decoder, using more effective way to exploit Kronecker products for factorized representations. Experimental comparisons with state-of-the-art model reduction approaches, not only PHM but also recently proposed DeLighT [14], on a range of machine translation problems, show that the approach offers similar translation quality with much fewer parameters.

## Funding Disclosure and Competing Interests

This work was partially funded by NSF grant IIS-1453658 to T.A.
The authors have no competing interests.

---

[4]Code available at: `https://github.com/tarodz/shapeshifter`.

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
