# OpenReview forum: "Shapeshifter: a Parameter-efficient Transformer using Factorized Reshaped Matrices"
_NeurIPS.cc/2021/Conference — NeurIPS 2021 Poster_

### Official Review · Reviewer_6iZJ · 2021-07-10

**Rating:** 6
**Confidence:** 3

**Summary:**

This paper proposes a method to reduce the number of parameters by weight factorization. Unlike traditional low-rank factorization (n x r and r x m), this work decomposes the matrix into two sqrt(mn) x r matrices. The algorithm includes the reordering of elements to recover the desired shape. The authors find connections to Kronecker products and theoretically support the expressiveness of the proposed method. Experiments on NMT tasks show that Shapeshifter can achieve better parameter efficiency (fewer parameters with comparable performance). The results are compared to two similar works: PHM and DeLighT.

**Ethical Concerns:**

There seems to be no ethical concern.

**Limitations And Societal Impact:**

The authors well addressed the limitations of theoretical analysis. Because the idea mainly focuses on efficiency, I do not think this paper needs more discussion about the social impact.

**Main Review:**

Overall, the idea is simple and efficient. The paper is well-written and easy to follow. I also appreciate that the authors explained experiments in detail and split the parameter savings for each component in every table.

I have a few concerns and suggestions:

(1) In section 5.3 and table 5, the expressiveness is showed by comparing with low-rank factorization. (“Despite more than two times… increased expressiveness of a stack of low-rank … layers” (line 340-343)) However, the parameter reduction almost comes from the embedding, so how can we know that different factorization in layers is the reason of better performance? I suggest using the same embedding factorization method (or no factorization for the embedding) to match the number of parameters and only change layers above for fair comparison. Also, there should be some explanation of the relationship between the number of parameters and expressiveness.

(2) I am not much familiar with the theoretical part, but does the analysis still holds after reshaping and transpose? Maybe these operations change the rank of the matrix?

(3) As mentioned, (line 156-157) I agree that Shapeshifter implementation should be faster than PHM (and possibly slower than naïve low-rank factorization). It will strengthen the paper if there are some inference speed comparisons.

(4) I am just curious that low-rank factorization (both naïve and Shapeshifter) ‘from-scratch’ is better than ‘decompose-after-training’. If we do not think about the training cost and the latter performs better, why should we train with a factorized form?

Minor issue: in table 5, “Ro-En” should be changed to “En-Ro”.

-------------
After reading the authors' responses and other reviews, I think my concerns are well addressed. I appreciate authors, especially for extra experiments, and I'll change my score from 5 to 6.

**Time Spent Reviewing:**

5

---

> ### Author Response · Authors · 2021-08-10
> **Response to Reviewer Comments**
>
> **Re: (1) – Comparison with standard low-rank representation**: We have performed a comparison between a rank-16 Shapeshifter and a rank-16 standard low-rank factorized model, with no compression of the input/output embedding matrix. The goal is to measure the expressivity of the core part of the Transformer model (attention and feed-forward layers) without impact from compressing the in/out embedding. The results are similar to what we reported in Table 5, that is, Shapeshifter outperforms the standard low-rank representation, both in terms of loss on the training set, and performance (loss and BLEU) on the dev set.
>
> Dataset: De-En
> * Train Loss:
>    * Shapeshifter: 2.66
>    * Low-rank: 2.86
> * Dev Loss:
>    * Shapeshifter: 2.73
>    * Low-rank: 2.90
> * BLEU
>    * Shapeshifter: 35.85
>    * Low-rank: 32.01
>
> Dataset: En-Ro
> * Train Loss:
>    * Shapeshifter: 2.16
>    * Low-rank: 2.33
> * Dev Loss:
>    * Shapeshifter: 3.09
>    * Low-rank: 3.34
> * BLEU
>    * Shapeshifter: 21.93
>    * Low-rank: 19.34
>
> In the models, as in the original Transformer model they are based on, the input embedding is tied to the output embedding, i.e., the linear transformation preceding the softmax layer that produces the final predictions uses the same matrix as the input word embeddings. The results show that not compressing the output layer (when embeddings are not compressed) has positive impact on the low-rank representation, but no impact on the Shapeshifter representation (in fact, scores are a little bit worse, most likely due to noise). This provides further evidence that a stack of Kronecker-based representations is more expressive than the low-rank representation: for the latter, providing an uncompressed, unconstrained layer at the top of the model instead of rank-256 decomposed matrix increases its expressivity as evidenced by higher BLUE and lower train/dev loss, while Shapeshifter is already expressive-enough with Kronecker-rank-256 matrix in the top layer. We will include these results and discussion in the updated manuscript.
>
>
> **Re: (2) – Effect of reshaping and transpose**: Indeed, these operations affect the rank of the matrix: in effect, instead of a rank-r matrix, we have a matrix that is equivalent to rank-r Kronecker product; this Kronecker-based decomposition, as we show, has higher expressiveness when stacked layer after layer than standard rank-r matrices.
>
> **Re: (3) – Speed comparisons**: We will expand the manuscript to add a paragraph about speed. The results below show that the method is only moderately slower (below 20% overhead) than the original, not compressed Transformer. To assess the overhead with as little interference from other jobs as possible, we trained the model on an isolated machine with a single NVIDIA RTX 3090 GPU, where the model was the only job running. We measured the forward/backward training time and forward-only test time for the original Transformer model, and for Shapeshifter 16/256 model (rank 16 for attention & feed-forward layers, rank 256 for in/out embedding matrix). We will include these results in the updated manuscript.
>
> Time to complete 10 epochs of training
> * Dataset: De-En
>    * Original Transformer: 2917 seconds
>    * Shapeshifter 16/256: 3426 sec. (17% increase)
> * Dataset: En-Ro
>    * Original Transformer: 16192 sec.
>    * Shapeshifter 16/256: 18007 sec. (11% increase)
>
> Time to perform predictions on the whole test set
> * Dataset: De-En
>    * Original Transformer: 203 sec.
>    * Shapeshifter 16/256: 219 sec. (8% increase)
> * Dataset: En-Ro
>    * Original Transformer: 110 sec.
>    * Shapeshifter 16/256: 124 sec. (13% increase)
>
>
> **Re: (4) – Why train from scratch?**: The reason for training from scratch is to avoid approximation errors resulting from approximating the matrices of the original model in a decomposed form – these errors add up in a multi-layer model such as a Transformer.

---

### Official Review · Reviewer_uuLV · 2021-07-14

**Rating:** 7
**Confidence:** 3

**Summary:**

This paper proposes an algorithm they call ‘Shapeshifter’ to reduce the number of parameters needed for Transformer-based models while preserving expressiveness and performance. The proposed algorithm is based on the idea of low-rank factorizations of weight matrices, but uses a different factorization method involving sums of Kronecker products that leads to greater expressiveness. They show theoretical analysis to prove their factorization can accurately represent any $n \times m$ matrix or stack of matrices (in an idealized case), and experimental results using transformer-based models for translation on English-German, English-French and English-Romanian. They demonstrate superior or comparable performance with vastly fewer parameters compared to some other model compression techniques.

**Limitations And Societal Impact:**

Little to no discussion of either as far as I can tell. It might be a good idea to add some discussion about the situations where this approach is and is not suitable compared to other model compression approaches.

**Main Review:**

This paper utilizes applies concepts proposed in previous work in a novel way to create a general method for reducing parameters needed in a wide variety of deep architectures. The proposed method is far more general than the work that came before it, and the reduction in parameter size it offers is applicable throughout almost any application in deep learning. I think it is a very strong paper overall, and a very useful contribution to the field.


The main criticisms I would make of this paper relate to the results and related work. First, I think it is odd that knowledge distillation methods such as DistilBERT are mentioned in the introduction, but not discussed in the related work or compared against in the results. I do understand that the proposed method is somewhat orthogonal to knowledge distillation and could perhaps even be used in concert with it. Nevertheless, the majority of successes in minimizing the size of transformer models in the literature thus far have used knowledge distillation (DistilBERT, TinyBERT, etc…), and a simple comparison of your method’s effectiveness against these existing approaches in preserving accuracy while reducing parameters would help make a stronger claim for the method’s usefulness. Without this comparison it is hard to judge if this using this method would lead to an actual improvement over existing techniques for shrinking model size. My one hesitation about giving this paper a higher score is due to my uncertainty over how it compares to these models.


Second, I think the results section could be better structured. There are 5 tables containing results from 5 different datasets, but multiple tables contain results from the same datasets and there are multiple different datasets used for the same language pairs. Many of the tables also lack captions and in places it is unclear how to interpret them (e.g. why are the results with the DeLighT architecture and PHM architecture both in table 2, with no explanation other than a horizontal line about what the difference between the first set of results and the second set of results is). These aren’t critical issues, but I found that I had to read through the results section three or four times before I understand what each of the tables was supposed to be representing.


Finally, I had one or two questions I wanted to ask about the architecture. First, at the end of section 4 you claim “Compared to the equivalent sum of Kronecker-products representation, using the matrix factorization representation avoids the explicit summation of terms, resulting in simpler, more efficient implementations”. Is it actually faster? It seems like it should have the same number of operations to me. Second, you mention that the $m \times n$ matrix is reshaped into a $\sqrt{mn} \times \sqrt{mn}$ matrix. Do you discuss how you handle the case where $\sqrt{mn}$ is not an integer?


**Time Spent Reviewing:**

4

---

> ### Author Response · Authors · 2021-08-10
> **Response to Reviewer Comments**
>
> **Comparison with knowledge distillation**: We are aware of several distillation-based models that focus on the same tasks as we do and would be directly comparable, however, unlike distilled variants of BERT or GPT-2, the focus of the authors of these models is on techniques for increasing model quality instead of on reducing the parameter size. For example, a recent paper [1] that uses knowledge distillation for sequence-to-sequence models. For the WMT’14 En-De task (the same task we report in Table 4), they show an increase in BLEU from 27.3 for the original Transformer to between 28.14-28.57 depending on the additional techniques used in training. The student model is the same size as the teacher model, though; there is no reduction in parameter count. Similarly, [2] shows increase BLEU scores with the same parameter count. It would indeed be interesting to see if these techniques work well for the reduced-parameter Shapeshifter models – however, our goal was to keep the Transformer baseline and our model as simple as possible, in order to provide clear view of the effects of using the reduced-size representation.
>
> [1] Wang, F., Yan, J., Meng, F., & Zhou, J. Selective Knowledge Distillation for Neural Machine Translation. arXiv preprint arXiv:2105.12967. ACL'2021
>
> [2] Li, B., Wang, Z., Liu, H., Du, Q., Xiao, T., Zhang, C., & Zhu, J. Learning light-weight translation models from deep transformer. arXiv preprint arXiv:2012.13866. AAAI'2021
>
> **Presentation of results**: We apologize for the confusion the layout of the tables has caused. We will revise the manuscript and extend the experimental section to include a figure summarizing the results, and we will separate the tables containing the DeLighT comparisons from PHM comparisons.
>
> **Architectural issues**: We will edit the manuscript to clarify these two issues:
> 1) Regarding the comparison with sum of Kronecker-product representation: indeed, in principle, the number of operations should be the same. The improved efficiency refers to readily-available high-level implementations, e.g. the torch.kron recently introduced in pytorch 1.8 would need to be followed by an explicit summation over the rank dimension, while simple matrix multiplication hides the summation over the rank dimension.
>
> 2) When square root is not an integer, we use the ceiling, and then only use a submatrix of the resulting slightly-too-large reconstructed matrix.

---

> > ### Comment · Reviewer_uuLV · 2021-08-24
> > **Thank you for the response**
> >
> > Thank you for the response, and for answering my questions about the architecture. I stand by my score.

---

### Official Review · Reviewer_UTjx · 2021-07-16

**Rating:** 6
**Confidence:** 3

**Summary:**

Shapeshifter reparameterizes dense kernels into a product of two lower-rank matrices. Shapeshifter introduces reshapes and transposes to make this reparameterization more efficient. The paper provides a theoretical justification for its expressivity and strong experimental results in machine translation.

**Limitations And Societal Impact:**

I don't believe there are any potential negative societal impact.

**Main Review:**

Originality: I haven't seen this exact factorization in previous literature, so I believe the idea is novel. It's a relative straightforward extension of low-rank factorization. I consider the simplicity a positive, however. TensorNetwork (https://github.com/google/TensorNetwork/tree/master/tensornetwork/tn_keras) is another generalization of low-rank factorization that I would have liked to see a comparison to.

Quality: Claims are supported with theoretical justification and experimental results. The decomposition into Kronecker products provides good intuition. As the authors note, "the [theoretical] results are limited to idealized multilayer models with linear activation". Given that I hoped to see more experimental results. Maybe at least language modeling and compare against BERT (GLUE, SQuAD)? Some practical matters were left unaddressed. What's the effect on the reshapes and transposes on training and inference speed? It's common in larger models to parallelize the dense layer across multiple accelerators. Is that possible here?

Clarity: Paper clearly explains the ideas and experiments in a way that would be easy to reproduce.

Significance: Given the ease of implementation, I could see this method attracting widespread use as a way to reduce parameter count. More experimental results and benchmarks are needed and above practical matters would need to be addressed for researchers and practitioners to feel confident in using this technique, however.

**Time Spent Reviewing:**

2

---

> ### Author Response · Authors · 2021-08-10
> **Response to Reviewer Comments**
>
> **Comparison with TensorNetworks**: We will extend the discussion of Tensorized Transformer [ref. 25 in our manuscript] – they performed tests on WMT’16 En-De machine translation, achieving BLEU scores similar to the original Transformer, with about 2.5-fold reduction in model size (52M parameters to 21M parameters). Our model on IWSLT’14 En-De dataset achieves more than 14-fold reduction in model size (from 60M to 4.2M parameters), while also achieving BLEU scores similar to the original Transformer, and on WMT’14 En-De dataset achieves 5.7-fold reduction (from 62M to 10.7M parameters) with a BLEU drop from 27.3 for the original Transformer to 26.6 for our reduced-size model.
>
> **More experimental results**: With the limited time available for response, we were not able to apply the method to BERT pretraining and then finetuning for GLUE and SQuAD tasks.
>
> **Effect on training and inference speed**: The time penalty is moderate, below 20% increase in running time. To assess it with as little interference from other jobs as possible, we trained the model on an isolated machine with a single NVIDIA RTX 3090 GPU, where the model was the only job running. We measured the forward/backward training time and forward-only test time for the original Transformer model, and for Shapeshifter 16/256 model (rank 16 for attention & feed-forward layers, rank 256 for in/out embedding matrix). We will include these results in the updated manuscript.
>
> Time to complete 10 epochs of training
> * Dataset: De-En
>    * Original Transformer: 2917 seconds
>    * Shapeshifter 16/256: 3426 sec. (17% increase)
> * Dataset: En-Ro
>    * Original Transformer: 16192 sec.
>    * Shapeshifter 16/256: 18007 sec. (11% increase)
>
> Time to perform predictions on the whole test set
> * Dataset: De-En
>    * Original Transformer: 203 sec.
>    * Shapeshifter 16/256: 219 sec. (8% increase)
> * Dataset: En-Ro
>    * Original Transformer: 110 sec.
>    * Shapeshifter 16/256: 124 sec. (13% increase)
>
> **Possibility of parallelization**: Introduction of the reduced-parameter representations does not affect the ability to do data parallelism by running minibatches on multiple GPUs and accumulating the gradients (in fact, while our experiments were on a single node, we used the LAMB optimizer that is common in large-batch parallel training). Each node working on its samples will introduce the overhead from the operations involved in the matrix multiplication / Kronecker product.

---

> > ### Comment · Reviewer_UTjx · 2021-08-23
> > **Response to authors**
> >
> > Thank you for addressing my concerns. I will raise my rating to 6.

---

### Official Review · Reviewer_tnWM · 2021-07-16

**Rating:** 7
**Confidence:** 4

**Summary:**

A sizable effort has been made in recent history to obtain the accuracy of very large language models at a fraction of the parameter of the large models. Broadly these methods fall into various categories like model compression, factorized representation, distillation etc. This paper underlines another such attempt at obtaining a factorized representation of matrices to reduce the parameter space. The authors justify the use of Kronecker products to factorize parameter space, by showing that stacked layers of low rank matrices increase expressiveness. They also show the effectiveness of their proposed method by achieving a 4 to 14-fold reduction in parameters without sacrificing accuracy.

**Limitations And Societal Impact:**

The authors have not discussed any adverse societal impact of their research, and justified it as they are trying to reduce the size of large language model, which should provide solutions to some of the limitations of large language models.

**Main Review:**

Contributions:
One of the key contributions of this paper is showing how unlike low rank matrix factorization, stacking multiple layers of Kronecker factors, has an effect on the kind of matrices that can be represented. In the process they prove / make the following key observation:

-- An orthogonal matrix can be  represented as a product of O(n^2) 2-variants givens matrices (known)

-- Any 2-variant Givens matrix can be represented as a product of at most three component matrices where each component matrix is a sum of two pairs of Kronecker products. (theorem 2).

-- This result extends to any n-by-m matrix in general.

-- If instead one uses the sum of r Kronecker products of  matric pairs then the number of such matrices that are needed to represent an orthogonal matrix is O(n^2/r).

-- Multiple experiments show that as compared to other factored representations such as low-rank, PHM the shapeshifter method achieves a better tradeoff in terms of parameter and BLEU score.

Weakness

-- Using Kronecker products to reduce/factorize the parameters has been explored in multiple works (and has been cited by the authors), thus reducing the novelty of shapeshifter.



Questions for the Authors:

-- Given that in line 144 -145 the multiplication step uses many operations such as transpose and reshape, it would have been good to see on synthetic examples the tradeoff between memory (perhaps parameter size) and time for shapeshifter and other baselines. I would assume that a lower parameter representation is achieved at the cost of higher computational requirements.

-- The previous question also points towards another relationship i.e. between computational cost (training time) and parameter reduction. Have the authors done any experiments on it or given it any thoughts?

Minor points

-- Instead of the result table, the tradeoff would be better visible with a plot of parameter size and BLUE score.


**Time Spent Reviewing:**

11

---

> ### Author Response · Authors · 2021-08-10
> **Response to Reviewer Comments**
>
> **Running time vs parameter size tradeoff**: The number of parameters in the reduced-size models is proportional to rank of the Kronecker-product representation. Thus, lower parameter count should actually lead to slightly lower computational cost, since lower rank translates to fewer floating-point operations. In practice, the introduction of the code to calculate the low-rank Kronecker product representation introduces overhead (we have observed moderate, below-20%, increase in running time of training & predictions, see detailed data below), but we did not observe much difference in the amount of the overhead across practically-relevant ranges of rank (e.g., attention/feed-forward layers rank in range 16-64).
>
> Time to complete 10 epochs of training
> * Dataset: De-En
>    * Original Transformer: 2917 seconds
>    * Shapeshifter 16/256: 3426 sec. (17% increase)
> * Dataset: En-Ro
>    * Original Transformer: 16192 sec.
>    * Shapeshifter 16/256: 18007 sec. (11% increase)
>
> Time to perform predictions on the whole test set
> * Dataset: De-En
>    * Original Transformer: 203 sec.
>    * Shapeshifter 16/256: 219 sec. (8% increase)
> * Dataset: En-Ro
>    * Original Transformer: 110 sec.
>    * Shapeshifter 16/256: 124 sec. (13% increase)
>
>
> **Presentation of results**: We will update the manuscript to include a figure showing performance vs. model size in addition to the tables.

---

> > ### Comment · Reviewer_S3X5 · 2021-08-21
> > **Response to rebuttal**
> >
> > Thanks for your response. I will stick to my rating :)

---

> > ### Comment · Reviewer_tnWM · 2021-08-31
> > **Rebuttal Response**
> >
> > Thanks for the timing experiments. I stand by my score.

---

### Official Review · Reviewer_7r5Z · 2021-07-17

**Rating:** 7
**Confidence:** 4

**Summary:**

This paper proposes Shapeshifter, a parameter efficient Transformer using low-ranked factorized representations and some reshaping tricks to achieve fast, space efficient and expressive linear layers.

**Main Review:**

I think this paper is a great contribution to existing literature that tries to make Transformers more parameter efficient.

Results look great and promising, the method also looks interesting and simple (in a good way).

The results look very good, and Shapeshifter models seem to be able to substantially compress Transformers by almost 20 times in some cases while maintaining competitive performance. Results look very promising. The method also looks simple enough to have widespread potential impact.

A particular nit I had is about the complexity. I think there is no mention or evaluation of flops/throughput/speed of the model. I was wondering if the authors could report that. Parameter counts are only one aspect of efficiency and it would be good to know how fast/slow the model is.

Another concern is also that experimental comparisons are quite limited and only conducted on MT tasks. It would be great if authors can provide more evidence that this works on another task other than MT alone.

Overall, I like the idea and i think this could be pretty impactful.



**Time Spent Reviewing:**

1.0

---

> ### Author Response · Authors · 2021-08-10
> **Response to Reviewer Comments**
>
> **Computational cost**: The time penalty for using the reduced-size model is moderate – below 20% increase in training / prediction time. To assess the computational overhead with as little interference from other jobs as possible, we trained the model on an isolated machine with a single NVIDIA RTX 3090 GPU, where the model was the only job running. We measured the forward/backward training time and forward-only test time for the original Transformer model, and for Shapeshifter 16/256 model (rank 16 for attention & feed-forward layers, rank 256 for in/out embedding matrix). We will include these results in the updated manuscript
>
> Time to complete 10 epochs of training
> * Dataset: De-En
>    * Original Transformer: 2917 seconds
>    * Shapeshifter 16/256: 3426 sec. (17% increase)
> * Dataset: En-Ro
>    * Original Transformer: 16192 sec.
>    * Shapeshifter 16/256: 18007 sec. (11% increase)
>
> Time to perform predictions on the whole test set
> * Dataset: De-En
>    * Original Transformer: 203 sec.
>    * Shapeshifter 16/256: 219 sec. (8% increase)
> * Dataset: En-Ro
>    * Original Transformer: 110 sec.
>    * Shapeshifter 16/256: 124 sec. (13% increase)
>
>
> **Other tasks**: Given the limited time for response, we did not manage to train the model on new datasets/new tasks.

---

### Official Review · Reviewer_S3X5 · 2021-07-23

**Rating:** 7
**Confidence:** 4

**Summary:**

This paper presents an approach called ShapeShifter to compress large language models. The paper follows the approach of reducing number of parameters by using factorized matrix representations. The primary idea is to represent the matrix as a sum of Kronecker products, instead of standard low-rank factorizations. The paper theoretically shows that for non-square matrices, the proposed approach allows for increased reduction in parameters for the same decomposition rank. For square matrices, it allows for increased expressiveness of the decomposition for the same rank. Experimental results on machine translation tasks show that the proposed approach performs much better than the state-of-the-art model compression approaches and is just slightly worse than the large models.

**Limitations And Societal Impact:**

1) Since the proposed compression still lags behind the large model it would be interesting to see some analysis on what exactly is this model not able to express, for instance how well does a rank r Kronecker product matrix represent a matrix from the original large model.
2) The paper does not talk about any reduction in training time or prediction time, it would be good to see some discussion on that as well.

**Main Review:**

The paper is very well written and easy to understand. The primary idea itself is quite novel, although the use of Kronecker products as a matrix factorization technique has been around for a long time but this paper shows its application in compression large neural network which is quite interesting.  The authors also provide ample theoretical and experimental evidence to show that their method is indeed better than the state-of-the-art.

**Time Spent Reviewing:**

2

---

> ### Author Response · Authors · 2021-08-10
> **Response to Reviewer Comments**
>
> **Re: 1 – Comparing with original model matrices**: Our models are trained to find a set of matrices in the Kronecker product representation that together work well as a stack of layers, starting from a random initialization. Because of this, individual matrices are not approximations of the matrices from the original large model, making comparisons using e.g. a matrix norm difficult.
>
> **Re: 2 – Computational cost**: In terms of training and prediction time, the reduced-size representation involving two smaller matrices comes at a cost of having to perform the multiplication of elements from these matrices. The cost is moderate (below 20%). To assess it with as little interference from other jobs as possible, we trained the model on an isolated machine with a single NVIDIA RTX 3090 GPU, where the model was the only job running. We measured the forward/backward training time and forward-only test time for the original Transformer model, and for Shapeshifter 16/256 model (rank 16 for attention & feed-forward layers, rank 256 for in/out embedding matrix).
>
> Time to complete 10 epochs of training
> * Dataset: De-En
>    * Original Transformer: 2917 seconds
>    * Shapeshifter 16/256: 3426 sec. (17% increase)
> * Dataset: En-Ro
>    * Original Transformer: 16192 sec.
>    * Shapeshifter 16/256: 18007 sec. (11% increase)
>
> Time to perform predictions on the whole test set
> * Dataset: De-En
>    * Original Transformer: 203 sec.
>    * Shapeshifter 16/256: 219 sec. (8% increase)
> * Dataset: En-Ro
>    * Original Transformer: 110 sec.
>    * Shapeshifter 16/256: 124 sec. (13% increase)

---

### Decision · Program_Chairs · 2021-09-27

**Decision:**

Accept (Poster)

**Comment:**

The paper attempts to improve parameter efficiency of transformer models for sequence modelling. In this regard, the authors propose a novel replacement of all matrices in the transformer by sum of Kronecker products (a type of low-ranked factorized representations) and some reshaping tricks. Further, the expressivity of such Kronecker-based linear layers is analysed. Empirical results on translation looks promising (better model-size vs performance trade-off as compared to other compression approaches for transformers). We thank the reviewers and authors for engaging in an active discussion, which resulted in clearing a lot of the concerns (e.g. speed/flops) and a lot of constructive feedback were provided to improve the paper. The authors provided new empirical results as part of the discussion, please include them in the final version of the paper as they add great value and understanding to the model as a whole.

- Please remove "fast" from abstract in the final version as the proposed method is slower than baseline in both training and inference.